# Quantifying the Impacts of Economic Progress, Economic Structure, Urbanization Process, and Number of Vehicles on PM_2.5_ Concentration: A Provincial Panel Data Model Analysis of China

**DOI:** 10.3390/ijerph16162926

**Published:** 2019-08-15

**Authors:** Haoran Zhao, Sen Guo, Huiru Zhao

**Affiliations:** 1School of Economics and Management, North China Electric Power University, Beijing 102206, China; 2Beijing Key Laboratory of New Energy and Low-Carbon Development, North China Electric Power University, Changping, Beijing 102206, China

**Keywords:** PM_2.5_ concentrations, GDP per capita, economic structure, urbanization rate, civil vehicles amount, panel data model

## Abstract

With the rapid development of China’s economy, the environmental problems are becoming increasingly prominent, especially the PM_2.5_ (particulate matter with diameter smaller than 2.5 μm) concentrations that have exerted adverse influences on human health. Considering the fact that PM_2.5_ concentrations are mainly caused by anthropogenic activities, this paper selected economic growth, economic structure, urbanization, and the number of civil vehicles as the primary factors and then explored the nexus between those variables and PM_2.5_ concentrations by employing a panel data model for 31 Chinese provinces. The estimated model showed that: (1) the coefficients of the variables for provinces located in North, Central, and East China were larger than that of other provinces; (2) GDP per capita made the largest contribution to PM_2.5_ concentrations, while the number of civil vehicles made the least contribution; and (3) the higher the development level of a factor, the greater the contribution it makes to PM_2.5_ concentrations. It was also found that a bi-directional Granger causal nexus exists between PM_2.5_ concentrations and economic progress as well as between PM_2.5_ concentrations and the urbanization process for all provinces. Policy recommendations were finally obtained through empirical discussions, which include that provincial governments should adjust the economic and industrial development patterns, restrict immigration to intensive urban areas, decrease the successful proportion of vehicle licenses, and promote electric vehicles as a substitute to petrol vehicles.

## 1. Introduction

With the rapid development of China’s economy, people’s living standards have largely been improved. Meanwhile, serious environmental problems have been triggered [1,2], especially that of atmospheric pollution. Climate change, haze, and fog weather have gradually gained the attention of people [3,4]. PM_2.5_ (particulate matter with a diameter smaller than 2.5 μm) has been deemed as the main constituent of haze and fog weather, which can threaten human health because it can be inhaled into the lungs [5,6]. Considering the detriments of PM_2.5_ on human beings and the negative influences on the environment, the central government of China has the goal to decrease PM_2.5_ concentrations to 35 μg/m^3^ in 2030, which was defined as the standard during the ‘transition period’ by the World Health Organization (WHO). To realize this goal, investigations into the influences of anthropogenic factors on PM_2.5_ concentrations are particularly significant [7,8]. However, only a few studies have quantitatively analyzed the impact mechanism of anthropogenic factors on PM_2.5_ concentrations including economic development, urbanization progress, economic structure, and population, which imply that the dynamic nexus among these factors are not well understood [9]. A better understanding of the complex nexus between those factors can help to discover the most significant anthropogenic factor on PM_2.5_ concentrations and the contribution degree of each anthropogenic factor to PM_2.5_ concentrations, which are critical for policy formulators to develop effective policies to reduce atmospheric pollution.

Considering that the monitoring of PM_2.5_ concentrations started around 2013 in China, the scarcity of long-range historical data for PM_2.5_ concentrations has brought about a shortage of studies that have explored the nexus between anthropogenic factors and PM_2.5_ concentrations. Therefore, the primary contributions of this paper include:

(1) Integrating the PM_2.5_ concentrations estimated through remote sensing [10] and data from several significant anthropogenic factors, then quantitatively analyzing the long-run relationship between these variables by utilizing a panel data theoretical framework for China’s 31 provinces, which include Pedroni co-integration examination and panel Granger causal nexus examination.

(2) Quantitatively exploring the complex relationship between these factors, so that policy implications can be suggested with regard to the various situations of different provinces, which are valuable references to provincial-level policy formulators.

The reminder of this paper is divided into six sections. The literature review is summarized in Section 2. The panel data theoretical framework is elaborated in Section 3. Section 4 describes the selection of independent variables and introduces the data sources. Section 5 presents the empirical analysis. Section 6 discusses the empirical results and proposes several pertinent policy suggestions. Our conclusions are presented in Section 7.

## 2. Literature Review

Several methodological frameworks and quantitative analyses have been employed to investigate the nexus among several socio-economic factors and atmospheric pollution. For example, the multi-objective method has been utilized to explore the interactions of economy, environment, and energy [11]. Moreover, the environmental Kuznets curve (EKC) assumption, combined with a different econometric methodology, was employed to investigate the coordinated relationship between economic development level, energy consumption, and environmental pollution [12]. Current studies have verified the existence of an inverse U-shape assumption of EKC between economic development and carbon dioxide (CO_2_) discharge [13,14,15,16,17,18,19], between economic progress and sulfur oxide (SO_2_) discharge [20,21,22], between economic development and nitrous oxide (NO_x_) discharge [23,24] as well as between economic progress and hazardous waste [25,26,27]. Furthermore, the system dynamics method has also been utilized to discuss the complicated nexus between the environment and the economic driving forces of the mining industry [28]. Additionally, the spatial econometric method [29,30,31,32] and panel data model [33] were also applied to analyze the nexus between environmental pollution and socio-economic factors.

Among the models summarized above, econometric models such as the panel data model have been extensively employed in analyzing the relationship between environmental pollution and economic driving forces. However, when compared with the large amount of research on CO_2_ emissions, SO_2_ discharge, and NO_x_ emissions, only a few works have studied the nexus between socio-economic drivers and PM_2.5_ concentration. The reasons for the scarcity of the literature in exploring the nexus between socio-economic drivers and PM_2.5_ concentrations are twofold. First, unlike conventional atmospheric pollutants such as SO_2_, NO_x_, and CO_2_, PM_2.5_ only threatens human health and the environment in developing countries or cities other than developed regions, hence only developing countries such as China, where haze and fog weather have frequently occurred in recent years, have paid much more attention to the PM_2.5_ topic. Second, with the remarkable attention given to PM_2.5_ recently, governments have started collecting PM_2.5_-related data in the last few years, hence the lack of long-range data of PM_2.5_ has overwhelmingly restricted the research in the area of PM_2.5_. Considering that PM_2.5_ concentrations have achieved a relatively high level in most of China’s provinces, the research on the nexus between anthropogenic factors and PM_2.5_ concentration is of great urgency for policy formulation to curb the PM_2.5_ issue. Several studies have employed satellite data of PM_2.5_ based on an econometric methodology to explore the influence of anthropogenic factors on PM_2.5_ concentrations. Li et al. [33] quantitatively analyzed the critical anthropogenic drivers responsible for the increase of PM_2.5_ concentrations in China by utilizing the panel data method from 1999 to 2011 at a city level. Xu and Lin [34] discussed the nexus among several significant driving factors and PM_2.5_ concentration at a regional level on the basis of the co-integration model. Hao and Liu [29] analyzed the social and economic influencing forces of the PM_2.5_ concentrations of China’s 73 cities in 2013 based on the spatial econometric method. Ding et al. [35] researched the relationship between economic development and PM_2.5_ pollution based on the spatial Durbin model by employing satellite observation data of PM_2.5_ pollution for 13 cities in the Beijing–Tianjin–Hebei region from 1998 to 2016. 

To fill the research gap in the complex relationship between anthropogenic forces and PM_2.5_ concentration, this paper analyzed the contribution of several significant anthropogenic factors to PM_2.5_ concentration and the causal relationship between these variables in 31 of China’s provinces. The critical contributions of this study are as follows. First, this research, for the first time to the best of our knowledge, analyzed the contributions of economic progress, urbanization, economic structure, and civil vehicles to PM_2.5_ concentration and the causality nexus between those variables based on panel data methodology from the provincial perspective in China; and second, based on the results of the empirical analysis, pertinent policy recommendations are provided with regard to the basic situations of various provinces which will make assist greatly in restricting the increase of PM_2.5_ concentrations. The results of the quantitative analysis in this paper can provide evidence for examining the long equilibrium nexus among anthropogenic forces and PM_2.5_ concentrations for 31 provinces in China, and the critical policy implications can provide significant references for future environmental conservation and the sustainable development of China.

## 3. Panel Data Methodology

This paper aimed to quantify the impacts of economic progress, urbanization, economic structure, and civil vehicles on PM_2.5_ concentrations for 31 Chinese provinces by employing the panel data methodology, which was first introduced into econometrics by Balestra [36]. In this research, the long-run relationship between these variables is elaborated below:(1)$$\ln PM_{2.5it}=\alpha +\beta_{1}\ln GDPPC_{it}+\beta_{2}\ln ur_{it}+\beta_{3}\ln es_{it}+\beta_{4}\ln ve_{it}+\varepsilon_{it}$$
where $PM_{2.5it}$ represents the PM_2.5_ concentrations; $GDPPC_{it}$ indicates the GDP per capita employed to represent economic development; $ur_{it}$ demonstrates the urbanization rate calculated by the proportion of urban population to total population; $es_{it}$ illustrates the economic structure indicated by the ratio of secondary industry added value to GDP; $ve_{it}$ implies the number of civil vehicles; $\varepsilon_{it}$ denotes the error component; i=1,2,…,31 are the researched provinces; t=1,2,…,T represents the time period; $\beta_{1},\beta_{2},\beta_{3}$ and $\beta_{4}$ are the elasticity coefficients of anthropogenic factors, respectively; and α is the constant term. All variables are in a natural logarithm form.

The panel data model can usually provide a large number of data points, thus the freedom degree of the data can be increased and the collinearity degree between the explanatory variables can be reduced. Meanwhile, the estimation effectiveness of the econometric model can be improved. However, the form and effectiveness of the panel data model need to be verified by several examinations. The elaborated panel data analysis procedures are as below:

Step 1: Panel data unit root examination

A precondition of the panel data analysis is to examine the stability of all data series. As commonly acknowledged, panel unit root examinations are more valid than unit root examinations on the basis of a univariate data sequence or cross sectional data sequences. This paper selected Levin, Lin and Chu (LL&C examination) [37] and Im, Pesaran, and Shin (IPS examination) [38] to examine the stability of all data series. Taking the AR (1) (auto-regression) procedure of the panel data into consideration:(2)$$y_{it}=\rho_{i}y_{it-1}+X_{it}\delta_{i}+\varepsilon_{it}$$
where $\rho_{i}$ implies the auto-regression coefficients; $X_{it}$ indicates the independent variables; and $\varepsilon_{it}$ demonstrates the error component. If $\left|\rho_{i}\right|<1$, then $y_{it}$ is deemed to be weakly stable. If $\left|\rho_{i}\right|=1$, then $y_{it}$ embodies a unit root [39]. 

Both the LL&C and IPS examination methods utilize the augmented Dickey–Fuller (ADF) specification as below:(3)$$\Delta y_{it}=\alpha_{i}y_{it-1}+\sum_{j=1}^{p_{i}}\beta_{ij}\Delta y_{it-j}+X_{it}^{\prime }\delta +\varepsilon_{it}$$
where Δ represents the first difference; $p_{i}$ demonstrates the lags number in regressions procedures; and $\alpha_{i}=\rho_{i}-1$ and $\Delta y_{it-j}$ illustrate the lag components ($j=1,2,\ldots ,p_{i}$).

Step 2: Panel co-integration examinations

Panel co-integration examinations are extensively acknowledged for their large capacity compared with the common time sequence co-integration [40,41]. The Pedroni panel co-integration examination [42] was utilized in this paper to examine whether there was a co-integration nexus between the selected variables. Unlike other conventional panel data methods, the Pedroni panel co-integration examination model admits trend coefficients and heterogeneous intercepts for cross-sections [42]. Such panel co-integration adheres to the form below:(4)$$y_{it}=\alpha_{i}+\delta_{i}t+\beta_{1i}x_{1i,t}+\beta_{2i}x_{2i,t}+\ldots +\beta_{Mi}x_{Mi,t}+e_{i,t}$$
where *M* indicates the amount of explanatory variables; $\beta_{1i},\ldots ,\beta_{Mi}$ represent the slope coefficients; and $\delta_{i}$ and $\alpha_{i}$ are the trend and individual effects, respectively. The null assumption of the Pedroni co-integration is that there is no co-integration, under which the residual $e_{i,t}$ in Equation (4) is integrated at one order. The null assumption should be rejected based on some statistics. Pedroni defines two kinds of examination statistics based on the residuals. One is related to the within dimension method that embodies the panel *ρ*-statistic, panel *v*-statistic, panel ADF-statistic, and panel PP-statistic. The other is the between dimension method, which contains the group PP-statistic, group *ρ*-statistic, and group ADF statistic.

Step 3: Examination of the model form

There are three critical forms, which are regression models, random effects as well as fixed effects models. The Hausman and the likelihood ratio (LR) test approach were utilized to examine the form of the established panel data model [43].

Additionally, the panel data approach contains fixed intercepts and the coefficients form (Equation (5)), varied intercepts and fixed coefficients form (Equation (6)), and the varied intercepts and coefficients form (Equation (7)) [44].
(5)$$y_{it}=\alpha +\beta x_{it}+\mu_{it}$$
(6)$$y_{it}=\alpha_{i}+\beta x_{it}+\mu_{it}$$
(7)$$y_{it}=\alpha_{i}+\beta_{i}x_{it}+\mu_{it}$$
where α indicates the intercept; *β* implies the coefficient; and $\mu_{it}$ illustrates the error term.

To choose an appropriate panel data model from Equations (5)–(7), an F-test was applied to determine whether the following null assumptions should be accepted via calculating the residual sum of squares (RSS) of Equations (5)–(7).
(8)$$\begin{array}{l} H_{1}:\beta_{1}=\beta_{2}=\cdots =\beta_{N}\\ F_{1}=\frac{\left(S_{2}-S_{1})/[(N-1)k\right]}{S_{1}/[NT-N(k+1)]}∼F((N-1)k,N(T-k-1)) \end{array}$$
(9)$$\begin{array}{l} H_{2}:\alpha_{1}=\alpha_{2}=\cdots =\alpha_{N},\beta_{1}=\beta_{2}=\cdots =\beta_{N}\\ F_{2}=\frac{\left(S_{3}-S_{1})/[(N-1)(k+1)\right]}{S_{1}/[NT-N(k+1)]}∼F((N-1)(k+1),N(T-k-1)) \end{array}$$
where *F*_1_ is calculated for testing the *H*_1_ assumption, which supposes that the coefficients are fixed and the intercepts are varied; *F*_2_ is computed to verify the *H*_2_ assumption, which sets that the coefficients and intercepts are fixed; and *S*_1_, *S*_2_, and *S*_3_ represent the *RSS* of Equations (5)–(7), respectively. *N*, *T*, and *k* demonstrate the number of researched provinces, time periods, and independent variables.

If the *F*_2_ value is smaller than the significant value, the *H*_2_ assumption can be accepted, then the panel data model is the type of Equation (5). If not, the *F*_1_ value needs to be calculated. If the *F*_1_ statistic is greater than the threshold value, the *H*_1_ assumption will be rejected, then the panel data model is in the form of Equation (7), or else, it is in the form of Equation (6).

Step 4: Granger causal nexus examination

The Granger causal nexus examination method [45] was utilized to examine the causality between PM_2.5_ concentrations, GDP per capita, economic structure, urbanization rate, and the number of civil vehicles. This approach was put forward by Engle and Granger [46], who verified that if two data series are co-integrated, a Granger causal nexus will exist between them. They also considered that if the forecasted values of Y were more accurate through utilizing the data of X and Y than that of only utilizing Y, it can be assumed that X Granger causes Y. The examination procedure of this approach can be written as:(10)$$y_{t}=\alpha +\sum_{i=1}^{m}\alpha_{i}y_{t-i}+\sum_{i=1}^{m}\beta_{i}x_{t-i}+e_{t}$$
(11)$$x_{t}=\alpha +\sum_{j=1}^{n}\alpha_{j}y_{t-j}+\sum_{j=1}^{n}\beta_{j}x_{t-j}+e_{t}$$

Equation (12) illustrates the null assumption of the Granger causal nexus examination approach, which means that ‘X does not Granger cause Y’, and Equation (13) is applied to examine if the Y Granger causes X.
(12)$$H_{0}:\beta_{i}=0,i=1,2,\ldots ,m$$
(13)$$H_{0}:\beta_{j}=0,j=1,2,\ldots ,n$$

## 4. Determining Independent Variables and Data Sources

This paper took four provincial level megacities (Beijing, Shanghai, Tianjin, and Chongqing), five autonomous regions, and 22 provinces as the research objects (termed as the 31 provinces for convenience). The PM_2.5_ concentration data utilized in this paper were estimated by integrating the data collected from the aerosol optical depth (AOD) for the moderate resolution imaging spectroradiometer (MODIS) of the National Aeronautics and Space Administration and the multi-angle imaging spectroradiometer (MISR) products imitated by the GEOS-Chem chemical conversion method [10,47]. It was verified that the satellite-based data were coherent with the ground-based data for China [10]. Therefore, we selected the PM_2.5_ concentration dataset [48] of the 31 Chinese provinces from 2000 to 2016 to analyze the relationship between anthropogenic forces and PM_2.5_ concentration.

The space distribution of the PM_2.5_ concentrations for 31 provinces in 2016 are illustrated in Figure 1. It can be seen that most provinces were much higher than 35 μg/m^3^, which is the goal set by the central government to achieve by 2030. The PM_2.5_ concentrations in the provinces located in central China, East China, North China and Xinjiang were higher than 50 μg/m^3^, which will pose great negative influences on human health and the environment. Considering the situation of PM_2.5_ concentrations in China, an analysis of the influences of anthropogenic forces on PM_2.5_ concentrations is of great urgency for policy formulation and sustainable development.

The significant anthropogenic forces selected in this paper are economic progress, urbanization rate, economic structure, and the number of civil vehicles. For economic progress, in light of the previous literature, it can be easily seen that the pollutant emissions are highly related with average income, which was utilized to represent economic progress [19,20,22]. Since several developed countries have experienced the emergence of haze and fog weather during the industrialization process, it is possible that the frequent occurrence of such weather is an essential process for China through the progress of the economy. Therefore, the development level of the economy can have great influences on PM_2.5_ concentrations. We selected GDP per capita to represent the economic progress, and the contribution of GDP per capita in the 31 provinces to PM_2.5_ concentrations was quantified.

For urbanization, previous studies have verified that the eco-environment was greatly influenced by the progress of urbanization and that atmospheric pollutants increased with speeding-up the process of urbanization [49,50,51,52]. Therefore, the urbanization process was also treated as a critical anthropogenic force of PM_2.5_.

For economic structure, since energy consumption intensive industries and pollutant discharging concentrated industries in secondary industry greatly contribute to various pollution emissions that threaten the environmental and atmospheric quality, the secondary industry in GDP was selected to demonstrate the economic structure and was taken as an explanatory variable.

For the number of civil vehicles, current investigations have identified that vehicle exhaust gas containing NO_x_, black carbon, and various pollutants are crucial sources of PM_2.5_ [53]. Therefore, the number of vehicles also exerts effects on PM_2.5_ concentration. In terms of the related available data, the number of civil vehicles was chosen as an explanatory variable.

The data of the above selected variables from 2000 to 2016 with regard to the 31 provinces were collected from the China Statistical Yearbook [54]. Due to the limited space, the descriptive statistics of these variables from 2014 to 2016 are listed in Table 1. The GDP per capita was converted into a constant price by taking 2000 as the fundamental period. All of the variables are in a natural logarithm form.

Table 2 illustrates the correlations between the PM_2.5_ concentration and the selected anthropogenic factors for the dataset of the panel data model. It indicates that the selected anthropogenic factors had a high correlation degree with PM_2.5_ concentration, which demonstrate that the economic development level, economic structure, urbanization, and number of civil vehicles play a significant role in the increase in PM_2.5_ concentration.

## 5. Empirical Analysis

This paper studied the relationship between PM_2.5_ concentration, economic progress, urbanization rate, economic structure, and the number of civil vehicles in 31 Chinese provinces by utilizing panel data methodology. The empirical analysis was processed as below.

**Step 1**: Examining the cross-sectional dependence

Since the panel data unit root examination approaches are divided into two categories: one that contains the LL&C examination [37] and IPS examination approaches [38], and the other embodies the approaches put forward by Bai and Ng [55], Moon and Perron [56], and Pesaran [57]. To determine the proper approaches, the cross-sectional dependence needed to be examined. The Pesaran examination method [58], proposed by Pesaran, was employed in this paper, and the results are listed in Table 3. As the *p*-value in Table 3 indicates, the null assumption was accepted at the 10% significance level. Hence, the approaches utilized to examine the unit root should not consider cross-sectional dependence.

**Step 2**: Examining the panel unit root

The approaches utilized to conduct panel unit root examinations do not need to take cross-sectional dependence into account, hence the LL&C test approach and IPS test approach were selected. As demonstrated in Table 4, in accordance with the probability statistics in the brackets of LL&C and IPS test results with regard to different variables, all variables were unstable in natural logarithm form. Then after first differencing, all variables were stationary as the probability statistics were smaller than the threshold values. Therefore, the PM_2.5_ concentrations and four explanatory variables were stationary after first differenced.

**Step 3**: Examining the panel co-integration

After confirming that all data series were stable after first differencing, we examined whether all data series were co-integrated before establishing the panel data model. Pedroni’s co-integration examination methodology was selected and the examining statistics are displayed in Table 5. In accordance with the probability values of various statistics, they were all smaller than the threshold value. Hence, it showed that there existed a long-run co-integration relationship between these variables.

Regarding the existence of structural breaks in the data series, we also employed the Westerlund panel co-integration test methodology, as proposed by Westerlund, to verify the evidence of the co-integration nexus among all variables. The details of the Westerlund panel co-integration test methodology can be referred to in [59]. The null hypothesis of the Westerlund panel co-integration test methodology is co-integration. According to the results listed in Table 5, the probability of the test statistic was 0.8971, which means that the null hypothesis of co-integration cannot be rejected. Therefore, it verified that a co-integration relationship exists between all variables.

**Step 4**: Identifying the model form

After verifying that a long-term co-integration relationship existed among all variables, we identified the model form to establish an appropriate panel data model. First, the random effect or fixed effect of model was examined by the LR and Hausman examination approaches, the results of which are illustrated in Table 6. The probability statistics of the LR examination were less than 1%, which demonstrates that the model should be fixed effect. The results of the cross-section random and probability statistics with regard to various variables in the Hausman examination also illustrate that the model was fixed effect.

Next, we needed to determine the model type from Equations (5)–(7), and the *F*-test was utilized to select the proper model type, and the results are presented in Table 7. To calculate the *F*-statistics, three RSS values of Equations (5)–(7) expressed by *S*_1_, *S*_2_, and *S*_3_ should first be obtained. Then, the *F*_1_ and *F*_2_ statistics can be computed in terms of Equations (8) and (9). After that, the model type can be identified by comparing the *F*_1_ and *F*_2_ statistics with the threshold values. If the *F*_2_ statistic is less than the threshold value *F*_2,*α*_*((N − 1)(K + 1),(NT − N(K + 1))*, the model can be written as Equation (5). If not, the *F*_1_ statistic needs to be examined. If the *F*_1_ statistic is greater than the threshold value *F*_1,*α*_*((N − 1)K,(NT − N(K + 1))*, the model can be expressed as Equation (7), otherwise, the model is in the form of Equation (6). Considering the results of the *F*-statistics, both the *F*_1_ and *F*_2_ statistics were greater than the threshold values at the supposed confidence level. Hence, the estimated model was a varied intercepts and coefficients model.

**Step 5**: Estimating the panel data model

In terms of the fundamental examinations above, the model was established based on the fixed effect with varied coefficients and intercepts taking PM_2.5_ concentration as the dependent variable, and GDP per capita, urbanization rate, economic structure, and the number of civil vehicles as explanatory variables for the 31 Chinese provinces. The evaluated coefficients and *t*-statistics listed in brackets are shown in Table 8. As implied from the *t*-statistics, all coefficients were significant at the 1% (represented by ^a^) or 5% (represented by ^b^) significance level. The *R*^2^ value was 0.9926, which indicates that the fitting effect of the established model was relatively high. The *F*-statistic was 131.86, higher than the threshold value, which indicates that the evaluated coefficients were significant. Therefore, the established model was verified to be significant and effective.

Since all of the variables are written in the logarithm form, the coefficients indicate elasticities which can demonstrate the contributions of various explanatory variables to PM_2.5_ concentration. Through comparatively analyzing the coefficients, we obtained several conclusions:

(1) Generally, the coefficients of the explanatory variables for the provinces in North China, Central China, and East China were much larger than that of the other provinces. Since some of the provinces in North China, Central China, and East China are megacities with a high urbanization rate and large population, and some of the provinces’ development depends on heavy industry and manufacturing industry, the selected anthropogenic factors were all deemed as the critical driving forces of high PM_2.5_ concentrations. The higher the level these factors realize, the greater contributions they will make to the increase in PM_2.5_ concentration. Therefore, compared with the relatively backward provinces located in Northwest China and Southwest China, the contributions of the anthropogenic factors of these provinces make greater contributions to the PM_2.5_ concentration.

(2) By comparing the coefficients of different variables of the corresponding provinces, it can be concluded that GDP per capita makes the largest contribution to PM_2.5_ concentration, the number of civil vehicles had the least impact on PM_2.5_ concentration, and the contribution degree of urbanization and economic structure relied on the urbanization progress and industrialization progress of different provinces. For Hebei, Inner Mongolia, Anhui, Henan, and Hunan, which have a relatively high proportion of secondary industry and a low level of urbanization, given that the economic development in these provinces depends highly on the secondary industry, thus leading to a high level pollutant emissions, the coefficients of the urbanization rate were less than that of economic structure. In contrast, for Beijing, Tianjin, Shanghai, and Guangdong, which have high levels of urbanization and a relatively large scale of tertiary industry, the contributions of urbanization in these provinces were much greater than that of industrialization on PM_2.5_ concentration.

(3) By comparing the coefficients of the same variable with regard to different provinces, it could be seen that when considering one anthropogenic factor, if the development of this factor for one province reaches a correspondingly high level, then the contribution of this anthropogenic factor to PM_2.5_ concentration would be relatively great. Taking GDP per capita as an example, the GDP per capita of Beijing, Tianjin, Shanghai, Jiangsu, and Zhejiang ranked the top five among the 31 provinces, and the contributions of GDP per capita of these five provinces to PM_2.5_ concentration also ranked in the top five when compared with other provinces.

**Step 6**: Examining the Granger causal nexus

The Ganger causal nexus examination results are illustrated in Figure 2. As can be seen in Figure 2, for all 31 provinces in China, the bi-directional Granger causal nexus existed between PM_2.5_ concentration and GDP per capita, and between PM_2.5_ concentration and urbanization rate. For Beijing, Tianjin, and Shanghai, there was a uni-directional causal nexus from economic structure to PM_2.5_ concentration. For Hainan and Tibet, there existed uni-directional causality from economic structure to PM_2.5_ concentration, and from the number of civil vehicles to PM_2.5_ concentration. For Qinghai and Ningxia, a uni-directional causality existed from the number of civil vehicles to PM_2.5_ concentration.

In terms of the Granger causal nexus examination results, it can be seen that:

(1) For all 31 provinces, a mutual influential relationship exists between PM_2.5_ concentration and economic development, and between PM_2.5_ concentration and urbanization, which implies that the accelerating development of economy and urbanization will lead to an increase in PM_2.5_ concentration, while a decrease in PM_2.5_ concentration will bring about the deceleration of economic development and urbanization progress.

(2) For provinces with a lower number of civil vehicles like Hainan, Tibet, Qinghai, and Ningxia as well as provinces with a small scale of secondary industry like Beijing, Tianjin, and Shanghai, a uni-directional causal nexus running from economic structure or the number of civil vehicles to PM_2.5_ concentration exists.

## 6. Discussion and Policy Recommendations

The existing literature has illustrated that the growth in PM_2.5_ concentration is critically influenced by anthropogenic forces [7]. Based on previous studies, this paper selected economic progress, economic structure, urbanization, and the number of civil vehicles as the critical anthropogenic factors that can reflect the developing tendency of the society and economy, and quantified the contributions of these factors to PM_2.5_ concentration by employing the panel data model based on the data from 31 provinces in China. The estimated coefficients indicate that by comparing the coefficients of different variables, the GDP per capita had the largest contribution to PM_2.5_ concentration, the number of civil vehicles had the least contribution to PM_2.5_ concentration, and the contributions of urbanization and economic structure relied on the urbanization progress and industrialization progress of different provinces. By comparing the coefficients of the same variable with regard to different provinces, we observed that if the development of the factor for one province reached a correspondingly high level, then the contribution of this anthropogenic factor to PM_2.5_ concentration would be relatively great. These results are different from a previous study [33] that used the panel data model to investigate the effect of economic growth, urbanization, and industrialization on PM_2.5_ concentration in China. In this study, industrialization was deemed as the most significant factor influencing PM_2.5_ concentration in industry-oriented, service-oriented, and heavily PM_2.5_ polluted provinces, and economic growth exerted more influence than the other factors on PM_2.5_ concentration in agriculture-oriented provinces. The Granger causal nexus examination results implied that a bi-directional causal relationship exists between PM_2.5_ concentration and economic development, and between PM_2.5_ concentration and urbanization. However, in [33], a bi-directional causal relationship existed between PM_2.5_ concentration and economic development in total panel, agriculture-oriented provinces, and heavily PM_2.5_ polluted provinces, and a bi-directional causal relationship existed between PM_2.5_ concentration and urbanization in the 31 provinces, except in heavily PM_2.5_ polluted provinces. Additionally, for provinces with a lower number of civil vehicles like Hainan, Tibet, Qinghai, and Ningxia as well as provinces with a small scale of secondary industry like Beijing, Tianjin, and Shanghai, a uni-directional causal nexus running from economic structure or the number of civil vehicles to PM_2.5_ concentration existed in these provinces. While in [33], a bi-directional causal relationship existed between PM_2.5_ concentration and economic structure in all provinces. Since econometrics methods are sensitive to data sequences, the differences in results are largely due to the inconsistent selection of variables and the range of the data as well as different regional divisions.

Considering the empirical analysis results, we can deduce that if China’s provinces maintain their current socio-economic developing pattern, the restriction of PM_2.5_ will exert undesirable influences on the progress of the economy, urbanization, and industrialization due to the bi-directional causality between these factors and PM_2.5_ concentration for most of the provinces in China. Therefore, policy makers need to make an appropriate balance of the restriction in the increase in PM_2.5_ and the progress of the economy, urbanization, and industrialization.

For the provinces with correspondingly large scales of secondary industry such as Hebei, Jilin, Inner Mongolia, Shanxi, Liaoning, Jiangsu, Anhui, Zhejiang, Shandong, Fujian, Jiangxi, Hubei, Henan, Guangdong, Hunan, Shaanxi, Guangxi, Ningxia, Qinghai, Chongqing, and Sichuan, of which their economic development primarily relies on the economic output of heavy and manufacturing industries, the contributions of economic structure in these provinces are relatively high. The high correlation between PM_2.5_ concentration, economic progress, and economic structure in these provinces urgently requires provincial governments to explore a novel developing pathway to adjust the economic development pattern from one pattern that consumes massive resources and energy with slow economic progress and a large amount of pollutants, to another pattern that relies on innovation technologies with less resource consumption, thus accelerating economic development and reducing pollution emissions. Furthermore, in consideration of the serious situation of the overcapacity of heavy industries leading to large pollutant emissions and the requirement of sustainable development, China’s industrial development pattern will have to be adjusted in the foreseeable future. Although the transformation of an industrial development pattern may be a difficult way, the following strategies to accelerate the transformation process can be taken: (1) provincial governments should encourage enterprises to increase investment into the research and development of high and novel technologies so that the industrial progress drivers can be converted from conventional resources and energy consumption to innovation and high technologies; (2) provincial governments should accelerate the elimination of backward production capacity; (3) the PM_2.5_ discharging standard of industrial enterprises should be improved to encourage enterprises to improve their pollutant treatment techniques; and (4) provincial governments should accelerate the adjustment of economic structure and prompt the development of the third industry.

For provinces with a high urbanization rate such as Beijing (achieving 86.50% in 2016), Tianjin (realizing 82.93% in 2016), and Shanghai (reaching 87.90% in 2016), the contributions of the urbanization rate of these provinces make greater contributions to the PM_2.5_ concentration when compared with provinces with a relatively low level of urbanization. Existing studies have demonstrated that cities with a correspondingly high level of urbanization usually have large PM_2.5_ concentrations [33]. Considering the urbanization process goal for China to achieve 60% in 2020 [60], we cannot decelerate the progress of urbanization to curb PM_2.5_ concentrations. Instead, provincial governments can restrict immigration to urban areas that are intensive and encourage residents to move into urban areas with low levels of urbanization so that the pressure of provinces with a high urbanization rate can be relieved, and the PM_2.5_ concentrations of these provinces can be restricted to increase. Additionally, the urbanization rate of backward provinces can be appropriately improved, so that the urbanization progress goal for the whole of China can be realized.

For provinces with a large number of civil vehicles, although the contribution of the number of civil vehicles was the least when compared with GDP per capita, economic structure, and urbanization rate, the contribution of the number of civil vehicles in provinces with a large a number of civil vehicles is greater than that of provinces with small numbers of civil vehicles. Since the exhaust gas from vehicles contains NO_x_, black carbon, and various pollutants that are critical sources of PM_2.5_, provincial governments should restrict the increase of vehicles to reduce PM_2.5_ and relieve traffic stress. Therefore, provincial governments need to control the proportion for vehicle licenses and promote the development of electric vehicles as a substitute for petrol vehicles.

## 7. Conclusions

With the accelerating development of China’s economy, urbanization and industrialization, the environmental problems are increasingly prominent. Atmospheric pollution, especially PM_2.5_, has gained wide attention from the population and governments given the adverse influences on human health. Since the growth of PM_2.5_ concentration is mainly caused by anthropogenic drivers, this paper selected several significant anthropogenic forces and explored the nexus between these factors and PM_2.5_ concentration to provide effective recommendations for policy makers to curb PM_2.5_. GDP per capita, economic structure, urbanization rate, and the number of civil vehicles were chosen as the explanatory variables to represent the social and economic situation of China. The panel data model was estimated by integrating these variables and taking PM_2.5_ concentration as the dependent variable based on the data of 31 Chinese provinces from 2000 to 2016. In light of the estimated coefficients in the model, we can conclude that: (1) In general, the coefficients of the explanatory variables for the provinces in North China, Central China, and East China were larger than that of the other provinces; (2) By comparing the coefficients of the different variables of the corresponding province, the GDP per capita made the largest contribution to PM_2.5_ concentration, the number of civil vehicles made the least contribution to PM_2.5_ concentration, and the contributions of urbanization and economic structure relied on the urbanization progress and industrialization progress; and (3) By comparing the coefficients of the same variable with regard to different provinces, it was seen that if the development of an anthropogenic factor of one province reached a correspondingly high level, then the contribution of this factor to PM_2.5_ concentration would be relatively great. In terms of the Granger causal nexus examination, we can conclude that: (1) bi-directional Granger causality exists between PM_2.5_ concentration and economic development, and between PM_2.5_ concentration and urbanization for all 31 provinces in China; and (2) for provinces with a lower number of civil vehicles like Hainan, Tibet, Qinghai, and Ningxia as well as provinces with a small scale of secondary industry like Beijing, Tianjin, and Shanghai, a uni-directional causal nexus from economic structure or the number of civil vehicles to PM_2.5_ concentration existed in these provinces.

In terms of the empirical discussion, the following policy recommendations are proposed:

(1) For provinces with a correspondingly large scale of secondary industry including Hebei, Shanxi, Inner Mongolia, and some other provinces, provincial governments are urgently required to explore a novel development pathway to adjust the economic development pattern from an extensive pattern of consumption of massive resources and energy with slow economic progress and large amounts of pollutants to an intensive pattern relying on innovation technologies with less resource consumption, thus accelerating economic development and reducing pollution emissions;

(2) To adjust the industrial development pattern for provinces with a large scale of secondary industry, the following strategies can be taken to accelerate the transformation process:Provincial governments should encourage enterprises to increase investment into the research and development of high and novel technologies so that the industrial progress drivers can be converted from conventional resources and energy consumption to innovation and high technologies;Provincial governments should accelerate the elimination of backward production capacity;The PM_2.5_ discharging standard of industrial enterprises should be improved to encourage enterprises to improve the pollutant treatment techniques; andProvincial governments should accelerate the adjustment of economic structure and prompt the development of tertiary industry.

(3) For provinces with a high urbanization rate such as Beijing, Tianjin, and Shanghai, provincial governments could restrict immigration to urban areas and encourage residents to move into urban areas with low levels of urbanization.

(4) For provinces with large numbers of civil vehicles, provincial governments need to control the proportion of vehicle licenses and promote the development of electric vehicles as a substitute for petrol vehicles.

It is worth noting that the results calculated based on future data may be quite different from the current results. This is partly due to the sensitiveness of econometric models to data sequences and partly to the different stages of China’s development. China, like other developed countries, will step into a new economic state, where people will no longer be willing to migrate to large crowded cities and heavy industries will no longer fuel economic growth, which means that economic growth will gradually depend on lighter industries, and finally, the services industries. The change in the economic state will result in a change in the causality nexus between PM_2.5_ concentration and other anthropogenic factors. Therefore, with the different development stages of China’s economy, we need to use the latest data sequences to explore the nexus between PM_2.5_ concentration and significant anthropogenic factors so that policy makers can formulate appropriate strategies to control the increase of PM_2.5_ concentrations.

One limitation of this investigation is that the empirical analysis was based on a provincial level, which is a large spatial scale. Different areas of a province may have various PM_2.5_ concentrations, particularly in rural areas and urban areas. However, it is difficult to obtain data of PM_2.5_ concentrations and anthropogenic factors for a panel of a refined spatial unit. Hence, if the data can be collected from PM_2.5_ observation stations, the study would be more helpful in future investigations. Moreover, future studies will focus on researching the causality nexus between PM_2.5_ concentration and other related factors such as exports and dust pollution. Some references [7,8] have provided evidence that exports and the development of construction industries, metal production sectors, and cement production sectors can exacerbate PM_2.5_ discharge.

## Figures and Tables

**Figure 1 ijerph-16-02926-f001:**
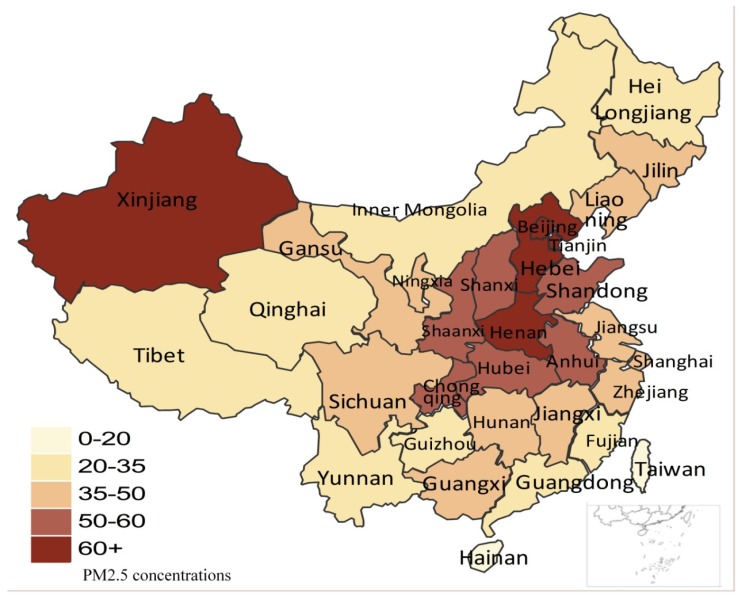
PM_2.5_ concentrations of 31 Chinese provinces in 2016.

**Figure 2 ijerph-16-02926-f002:**
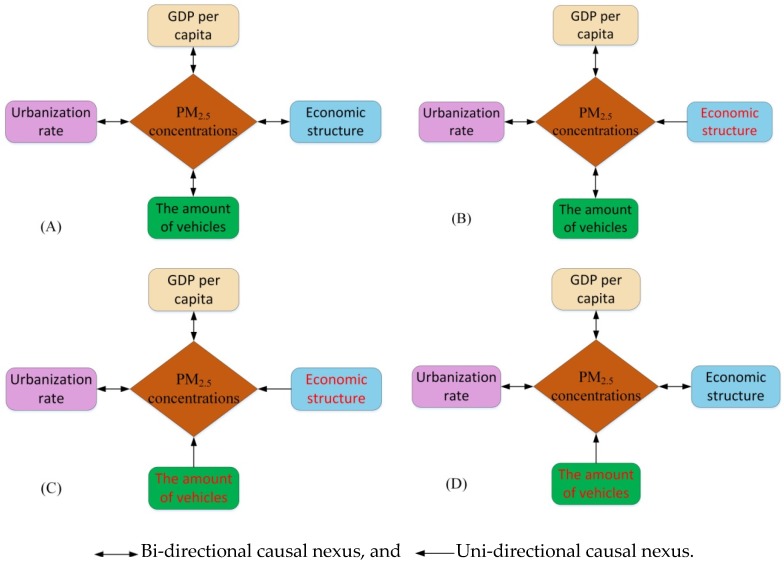
Granger causal nexus among all variables for the 31 provinces. (**B**) Beijing, Tianjin, and Shanghai, (**C**) Hainan and Tibet, (**D**) Qinghai and Ningxia, and (**A**) for the rest of the 31 provinces.

**Table 1 ijerph-16-02926-t001:** Descriptive statistics of parts of variables.

Variables	Units	2014			2015			2016		
		Mean	S.D.	M.V.	Mean	S.D.	M.V.	Mean	S.D.	M.V.
PM_2.5_	μg/m^3^	57.04	17.34	56.33	48.88	15.31	51.25	45.51	14.35	45.17
GDPPC	yuan/ren	50,742	21,721	40,648	54,727	23,355	44,225	58,783	25,034	47,586
ur	%	55.76	13.27	53.79	56.64	12.68	55.12	57.63	12.28	55.34
es	%	45.68	7.74	47.73	43.04	7.62	45.70	41.41	7.55	44.52
ve	10^4^	470.91	351.30	384.88	525.30	387.29	435.37	599.18	443.42	491.23

Note: S.D. means standard deviation and M.V. is the median value.

**Table 2 ijerph-16-02926-t002:** Correlations for the panel dataset.

Variables	PM_2.5_	GDPPC	ur	es	ve
PM_2.5_	1	0.985 *	0.897 *	0.853 *	0.792 *
GDPPC		1	0.886 *	0.802 *	0.603 *
ur			1	0.714 *	0.712 *
es				1	0.654 *
ve					1

Note: All variables are in a natural logarithm form. * demonstrates a 5% significance level.

**Table 3 ijerph-16-02926-t003:** Cross-sectional dependence examination results.

Cross-Sectional Dependence Examination	Pesaran’s Test	*p*-Value
Pesaran examination	2.1257	0.1016

**Table 4 ijerph-16-02926-t004:** Panel unit root examination results.

Form	Variables	L.L&C	IPS	Conclusions
Level	ln*PM*_2.5_	0.7638	0.5025	Non-stationary
		(0.7125)	(0.4623)	
	ln*GDPPC*	0.3657	1.5634	Non-stationary
		(0.6435)	(0.8547)	
	Ln*ur*	−0.5636	−0.8726	Non-stationary
		(0.1321)	(0.2821)	
	Ln*es*	1.7823	2.2531	Non-stationary
		(0.7238)	(0.8942)	
	Ln*ve*	2.7835	1.7629	Non-stationary
		(0.7968)	(0.8864)	
First Differenced	Δln*PM*_2.5_	−3.9043	−3.2764	Stationary
		(0.0186) ^b^	(0.0321) ^b^	
	Δln*GDPPC*	−3.4967	−3.9839	Stationary
		(0.0003) ^a^	(0.0121) ^b^	
	Δln*ur*	−3.9721	−3.4533	Stationary
		(0.0008) ^a^	(0.0011) ^a^	
	Δln*es*	−4.0125	−3.2045	Stationary
		(0.0001) ^a^	(0.0034) ^a^	
	Δln*ve*	−3.2636	−3.0695	Stationary
		(0.0025) ^a^	(0.0129) ^b^	

Notes: The values in brackets demonstrate the probability statistics. If the probability statistics are smaller than the threshold value, the data series are stable. ^a^ implies the 1% confidence level, and ^b^ indicates the 5% confidence level (which are the same for Table 5, Table 6 and Table 8).

**Table 5 ijerph-16-02926-t005:** Panel co-integration examination results.

Test Method	Test Statistics	Value	Probability
Pedroni	Panel v-Statistic	−2.7521	0.0217 ^b^
	Panel ρ-Statistic	−2.2341	0.0308 ^b^
	Panel PP-Statistic	−6.7238	0.0005 ^a^
	Panel ADF-Statistic	−2.0316	0.0201 ^b^
	Group ρ-Statistic	−2.3074	0.0103 ^b^
	Group PP-Statistic	−7.4567	0.0000 ^a^
	Group ADF-Statistic	−2.3519	0.0231 ^b^
**Westerlund**	**Panel LM test statistic**	**0.65**	**0.8971**

**Table 6 ijerph-16-02926-t006:** Panel data model effect identification results.

**LR Examination Results**		**Statistic**	**Prob.**	
Cross-section *F*		32.693647	0.0000 ^a^	
**Hausman Examination Results**				
	**Chi-Square Statistic**		**Prob.**	
**Cross-Section Random**	38.453127		0.0000 ^a^	
Variables	Fixed	Random	Var(Diff.)	Prob.
GDP per capita	0.515858	2.134375	0.134375	0.0000 ^a^
Urbanization rate	0.347378	1.372033	0.270118	0.0487 ^b^
Economic structure	0.448862	0.954043	0.049523	0.0232 ^b^
The number of civil vehicles	0.449145	0.951714	0.086230	0.0322 ^b^

**Table 7 ijerph-16-02926-t007:** *F*-test results.

*S* _1_	*S* _2_	*S* _3_	*F* _1_	*F* _2_
0.04372	0.69041	0.87660	7.64326	7.87508

**Table 8 ijerph-16-02926-t008:** The evaluated coefficients for the panel data model.

	Variables	*lnGDPPC*	*lnur*	*lnes*	*lnve*
Provinces					
North China	Beijing	0.1312	0.1243	0.0987	0.0767
		(4.2389) ^a^	(3.5256) ^b^	(4.0967) ^a^	(4.8048) ^a^
	Tianjin	0.1302	0.1198	0.0912	0.0699
		(4.0823) ^a^	(3.0278) ^b^	(3.8923) ^a^	(4.5623) ^a^
	Hebei	0.1218	0.0834	0.1064	0.0653
		(4.0174) ^a^	(3.6571) ^b^	(4.9937) ^a^	(3.8971) ^a^
	Shanxi	0.0912	0.0821	0.0812	0.0621
		(3.5467) ^b^	(3.8901) ^a^	(3.9937) ^a^	(3.4127) ^b^
	Inner Mongolia	0.1178	0.0701	0.0756	0.0598
		(3.4412) ^b^	(3.7029) ^b^	(3.6912) ^b^	(3.4567) ^b^
Northeast China	Liaoning	0.1134	0.0917	0.0859	0.0723
		(3.4268) ^b^	(3.7924) ^a^	(3.7529) ^b^	(3.9716) ^a^
	Jilin	0.1067	0.0851	0.0872	0.0662
		(3.3987) ^b^	(3.2496) ^b^	(3.8196) ^a^	(3.5917) ^b^
	Heilongjiang	0.0927	0.0819	0.0692	0.0701
		(3.4927) ^b^	(3.7624) ^a^	(3.7219) ^a^	(3.4103) ^b^
East China	Shanghai	0.1301	0.1299	0.0701	0.0512
		(3.8219) ^a^	(4.1716) ^a^	(3.8129) ^a^	(6.1714) ^a^
	Jiangsu	0.1288	0.0988	0.0867	0.0836
		(3.7918) ^a^	(4.0102) ^a^	(3.4186) ^b^	(4.5129) ^a^
	Zhejiang	0.1256	0.0964	0.0859	0.0825
		(3.6216) ^b^	(3.9921) ^a^	(3.6927) ^b^	(4.0321) ^a^
	Anhui	0.0889	0.0811	0.1102	0.0619
		(3.4356) ^b^	(3.7291) ^a^	(3.6927) ^a^	(3.4641) ^b^
	Fujian	0.1112	0.0976	0.0979	0.0561
		(3.5621) ^b^	(3.6215) ^b^	(3.5729) ^b^	(3.5743) ^b^
	Jiangxi	0.0877	0.0814	0.1083	0.0501
		(3.4218) ^b^	(4.0291) ^a^	(3.6420) ^b^	(3.9204) ^a^
	Shandong	0.1109	0.0902	0.0923	0.0868
		(3.6109) ^b^	(3.9413) ^a^	(3.7124) ^a^	(4.0371) ^a^
Central China	Henan	0.1217	0.0812	0.0843	0.0818
		(3.5192) ^b^	(3.4982) ^b^	(4.0327) ^a^	(3.9947) ^a^
	Hubei	0.1163	0.0859	0.0798	0.0721
		(3.6492) ^b^	(3.6132) ^b^	(3.7986) ^b^	(3.6125) ^b^
	Hunan	0.1098	0.0727	0.0782	0.0726
		(3.5617) ^b^	(3.4697) ^b^	(3.6961) ^b^	(3.8219) ^a^
South China	Guangdong	0.1068	0.1001	0.0767	0.0871
		(3.9031) ^a^	(3.6952) ^b^	(3.5829) ^b^	(4.2069) ^a^
	Guangxi	0.0801	0.0785	0.0782	0.0514
		(3.8129) ^a^	(3.7934) ^b^	(3.6027) ^b^	(4.0204) ^a^
	Hainan	0.0712	0.0602	0.0501	0.0329
		(3.8041) ^b^	(3.7549) ^b^	(3.5123) ^b^	(3.8024) ^b^
Northwest China	Shaanxi	0.1132	0.0835	0.1083	0.0753
		(4.0348) ^a^	(3.8157) ^a^	(3.8927) ^a^	(4.0128) ^a^
	Gansu	0.0757	0.0631	0.0746	0.0479
		(4.1279) ^a^	(3.6218) ^b^	(4.0182) ^a^	(3.9629) ^b^
	Qinghai	0.0769	0.0717	0.0859	0.0388
		(3.5938) ^b^	(4.0225) ^a^	(3.8864) ^a^	(4.0031) ^a^
	Ningxia	0.1071	0.0874	0.0821	0.0425
		(3.9082) ^b^	(3.9205) ^a^	(3.9764) ^a^	(3.7059) ^a^
	Xinjiang	0.1049	0.0789	0.0725	0.0633
		(4.5128) ^a^	(4.2981) ^a^	(3.8094) ^a^	(3.9421) ^a^
Southwest China	Chongqing	0.1147	0.0911	0.0767	0.0614
		(4.2571) ^a^	(4.4468) ^a^	(3.9647) ^a^	(3.8733) ^b^
	Sichuan	0.0908	0.0803	0.0808	0.0744
		(3.9044) ^b^	(4.3781) ^a^	(4.1163) ^a^	(3.9917) ^b^
	Guizhou	0.0745	0.0647	0.0733	0.0609
		(3.9256) ^a^	(3.9862) ^a^	(3.8022) ^b^	(3.6865) ^b^
	Yunnan	0.0724	0.065	0.0719	0.0715
		(4.0322) ^a^	(3.8874) ^b^	(3.7906) ^b^	(3.7162) ^b^
	Tibet	0.073	0.0401	0.0688	0.0299
		(4.1175) ^a^	(3.6549) ^b^	(3.5853) ^b^	(3.6104) ^b^

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
