# Peer review of "Quantifying the Impacts of Economic Progress, Economic Structure, Urbanization Process, and Number of Vehicles on PM2.5 Concentration: A Provincial Panel Data Model Analysis of China"

_ijerph, 2019, doi:10.3390/ijerph16162926_

Round 1

Reviewer 1 Report

Please, cite thie relvant and recent work: Magazzino, C., (2017), The relationship among economic growth, CO2 emissions, and energy use in the APEC countries: a panel VAR approach, Environment Systems and Decisions, 37, 3, 353-366.

Applied results should be discussed more in detail.

Policy implications must be strenghtened.

The English and the editing are poor.

Author Response

1. Please, cite this relevant and recent work: Magazzino, C., (2017), The relationship among economic growth, CO2 emissions, and energy use in the APEC countries: a panel VAR approach, Environment Systems and Decisions, 37, 3, 353-366.

Thank you very much for your comment and kind suggestion. We have cited this relevant and recent work in Literature Review section and marked it in yellow color in References section. Please check it in the manuscript.

2. Applied results should be discussed more in detail.

Thank you very much for your comment and suggestion. We have added some discussion about empirical results in Step 5 and Step 6 in Section 5 of the manuscript. Please check the added contents marked in yellow color in the manuscript.

3. Policy implications must be strengthened.

Thank you very much for your comment and suggestion. Policy implications are proposed based on the empirical analysis results including the strategies to accelerate the industrial development pattern transformation, the ways to decrease the influences of urbanization process on PM2.5 concentrations, and the ways to control the influences of the amount of civil vehicles on PM2.5 concentrations. We have revised some policy implications. Please check the revisions marked in yellow color in Section 6 and Section 7 in the manuscript.

4. The English and the editing are poor.

Thank you very much for your comment. We have revised the English style of this manuscript under the help of a professor working in the University of Michigan, Ann Arbor. Please check the revisions in the manuscript.

Reviewer 2 Report

This revised  manuscript is scientifically sound and can be accepted for publish.

Author Response

This revised manuscript is scientifically sound and can be accepted for publish.

Thank you very much for your kind comment.

This manuscript is a resubmission of an earlier submission. The following is a list of the peer review reports and author responses from that submission.

Round 1

Reviewer 1 Report

The authors successfully correlated the anthropogenic factors and PM2.5 concentrations with mathematical model and verified it with empirical data from China. The authors also did a fine job in processing the empirical analysis properly.

Authors should provide more inside information or references in discussion and policy recommendations section to support their findings. Readers can not realize and judge how the recommended policies work based on the results of this manuscript.

Overall, this manuscript is scientifically sound but its English writing fluency need to be improved to make this manuscript more readable.   

Author Response

Dear Editor,

Thank you very much for your work. Thanks a lot for the reviewers’ comments careful check, and their kind suggestions on our manuscript. We provide this cover letter to explain, point by point, the details of our revisions in the manuscript and our responses to the reviewers’ comments as follows. In order to make the changes easily viewable for you and reviewers, we marked the revisions in the revised manuscript in red color. We hope the revised manuscript would satisfy you and reviewers. We are looking forward to hearing from you soon. 

Best regards,

Sen Guo

1       Response to Reviewer 1

Revisions list according to the suggestions from Reviewer 1

The authors successfully correlated the anthropogenic factors and PM2.5 concentrations with mathematical model and verified it with empirical data from China. The authors also did a fine job in processing the empirical analysis properly.

1. Authors should provide more inside information or references in discussion and policy recommendations section to support their findings. Readers can not realize and judge how the recommended policies work based on the results of this manuscript.

Thank you very much for your comment and suggestion. We have added the contents about comparing results in this paper with previous results. We have also strengthened the policy implications based on the coefficients of panel data model and results of Granger causality examination. Please check the added contents and revisions marked in red color in Section 6 in the manuscript.

2. Overall, this manuscript is scientifically sound but its English writing fluency need to be improved to make this manuscript more readable.

Thank you very much for your comment and suggestion. The language of this manuscript has been revised by a professor from the University of Michigan, Ann Arbor. Please check the revisions in the manuscript.

Reviewer 2 Report

The topic of the paper has been largely debated in literature, and a great amount of studies has been devoted to this issue. The author should explain the choice of the case study, and its relevance. The originality aspects of the paper are not clearly discussed. The Introduction should better describe the topic, the applied analysis and the expected results, as well as the novelty aspects of the research.

The literature review is partial and incomplete, and some recent conttrbutions should be cited and discussed: i.e., Magazzino, C., Cerulli, G., (2019), The Determinants of CO2 Emissions in MENA Countries: A Responsiveness Scores Approach, International Journal of Sustainable Development & World Ecology, DOI: 10.1080/13504509.2019.1606863.

The model is described in a too simplistic way. Who introduced it? What about its limits, pros and cons? Descriptive statistics are totally missed. What about the values of mean, median, IQR, skewness, kurtosis, Standard Deviation and range? What about outliers and aberrant observations? A figure might improve the data description. Correlation matrix should be presented and discussed. 

Additional Cross-sectional panel dependence tests are needed.

Results of Table 2 are completely wrong: as shown by results in Table 1 cross-sectional dependence emerges, so second generation of panel unit root tests must be conducted instead of firs generation ones.

Additional panel cointegration tests are needed.

Comparisons with previous results and policy implications should be strengthen.

The English style is unacceptable: the paper must be re-written.

Author Response

Dear Editor,

Thank you very much for your work. Thanks a lot for the reviewers’ comments careful check, and their kind suggestions on our manuscript. We provide this cover letter to explain, point by point, the details of our revisions in the manuscript and our responses to the reviewers’ comments as follows. In order to make the changes easily viewable for you and reviewers, we marked the revisions in the revised manuscript in red color. We hope the revised manuscript would satisfy you and reviewers. We are looking forward to hearing from you soon. 

Best regards,

Sen Guo

 Response to Reviewer 2

Revisions list according to the suggestions from Reviewer 2

1. The topic of the paper has been largely debated in literature, and a great amount of studies has been devoted to this issue. The author should explain the choice of the case study, and its relevance. The originality aspects of the paper are not clearly discussed. The Introduction should better describe the topic, the applied analysis and the expected results, as well as the novelty aspects of the research.

Thank you very much for your comment and suggestion. With the dramatic speed of China’s economic progress, not only people’s living levels have been largely enhanced, but also seriously environmental problems have been triggered, especially the atmospheric pollution. Most of the existing literatures have contributed to the relationship among several socio-economic factors and carbon dioxide emissions, sulfur dioxide emissions or nitrogen oxide emissions. Considering the lack of PM2.5 concentrations data, the literatures focused on the relationship among socio-economic factors and PM2.5 concentrations are rarely. Since the haze and fog weather frequently happen in most provinces in China in recent years and PM2.5 has been deemed as the main constituent of haze and fog weather, central government of China aims at decreasing the PM2.5 concentrations to 35μg/m3 in 2030 to curb the serious and continuous PM2.5 pollutant. On the purpose of realizing this goal, investigating the influences of anthropogenic factors on PM2.5 is particularly significant. Therefore, this paper studies on the relationship among the anthropogenic factors and PM2.5 concentrations with econometrics model.

We have better described the topic, the applied analysis and expected results in Introduction section. We have also highlighted the novelty aspects of this research in Introduction section. Please check the revisions and added contents marked in red color in Introduction section in the manuscript.

2. The literature review is partial and incomplete, and some recent contributions should be cited and discussed: i.e., Magazzino, C., Cerulli, G., (2019), The Determinants of CO2 Emissions in MENA Countries: A Responsiveness Scores Approach, International Journal of Sustainable Development & World Ecology, DOI: 10.1080/13504509.2019.1606863.

Thank you very much for your comment and suggestion. We have added some recent literatures in Literature Review section to make it complete. Please check the added references in Literature Review section and References section marked in red color.

3. The model is described in a too simplistic way. Who introduced it? What about its limits, pros and cons? Descriptive statistics are totally missed. What about the values of mean, median, IQR, skewness, kurtosis, Standard Deviation and range? What about outliers and aberrant observations? A figure might improve the data description. Correlation matrix should be presented and discussed.

Thank you very much for your comment and suggestion. The panel data model was first introduced into econometrics by Balestra. Then with the development of econometrics, generally, the panel data model, employed to investigate the complicated relationship among several socio-economic factors and air pollutions, is consisted of several examination methods, including cross-sectional dependence test, panel unit root test, panel co-integration test, model form determination, and Granger causality test. Each test method was proposed by different people. The method used to examine cross-sectional dependence is proposed by Pesaran, the methods employed to examine panel unit root are proposed by Levin, Lin and Chu and Im, Pesaran and Shin, the method utilized to examine panel co-integration is proposed by Pedroni, the method used to determine the panel data model form is proposed by Hausman, and the method used to examine Granger causality nexus is proposed by Granger and Engle. These contents are implied in Section 3 and Section 5. Please check them in the manuscript.

The advantages and disadvantages of the panel data model have been added in the second paragraph in Section 3. Please check the added contents marked in red color in the manuscript.

We have added Table 1 to indicate the descriptive statistics of all selected variables. Please check the added contents marked in red color in Section 4 in the manuscript. We also use Figure 1 to illustrate the distribution of PM2.5 in 31 provinces in China. Please check these contents in Section 4 in the manuscript.

Table 2 is added to indicate correlations among PM2.5 concentrations and the selected anthropogenic factors for the panel data model data set. We have added discussion about correlation degree among variables in Section 4. Please check the added contents marked in red color in the manuscript.

4. Additional Cross-sectional panel dependence tests are needed.

Thank you very much for your comment and suggestion. Generally, cross-sectional dependence test is used to determine the panel data unit root examination approaches. In the existing literatures using panel data model to analyze the relationship among atmospheric pollutions and socio-economic factors, Pesaran examination method is a common method to conduct cross-sectional dependence. Therefore, this paper employed this testing method to examine cross-sectional dependence and the methods used to examine unit root are determined by the cross-sectional dependence test results.

5. Results of Table 2 are completely wrong: as shown by results in Table 1 cross-sectional dependence emerges, so second generation of panel unit root tests must be conducted instead of firs generation ones.

Thank you very much for your comment and suggestion. We have added second generation of panel unit root tests in Table 2. Please check the added contents marked in red color in Step 2 of Section 5 in the manuscript.

6. Additional panel cointegration tests are needed.

Thank you very much for your comment and suggestion. In the existing literatures investigating on the relationship among air pollutions and socio-economic factors employing panel data model, Pedroni’s co-integration examining methodology is a common method used to examine whether all variables are co-integrated as there are various test statistics used to judge the co-integration relationship among variables. Therefore, we select Pedroni’s co-integration examining methodology to testify there exists co-integration relationship among all variables so that the panel data model can be established.

7. Comparisons with previous results and policy implications should be strengthen.

Thank you very much for your comment and suggestion. We have added the contents about comparing results in this paper with previous results. We have also strengthened the policy implications based on the coefficients of panel data model and results of Granger causality examination. Please check the added contents and revisions marked in red color in Section 6 in the manuscript.

8. The English style is unacceptable: the paper must be re-written.

Thank you very much for your comment and suggestion. The language of this manuscript has been revised by a professor from the University of Michigan, Ann Arbor. Please check the revisions in the manuscript.

Round 2

Reviewer 2 Report

The Author did not understand my comments and disattended the econometric suggestions: i.e., I claimed the results for second generation of panel unit root tests, while they have shown results of first generation panel unit root tests on the second difference of the variable! This clearly shows the econometric inadequacy!

The English should be completely revised.

Policy implications are naive.